# Spatial Variation in the Frequency of Left-Sided Morph in European Flounder *Platichthys flesus* (Linnaeus, 1758) from the Marginal Arctic (the White Sea)

Peter N. Yershov [1,*], Gennadiy V. Fuks [2] and Vadim M. Khaitov [3,4]

1   Zoological Institute of the Russian Academy of Sciences, Universitetskaya nab. 1, 199034 Saint Petersburg, Russia
2   North Branch of "VNIRO" ("Severnyy"), Uritskogo st. 17, 163002 Arkhangelsk, Russia
3   Department of Invertebrate Zoology, Saint-Petersburg State University, Universitetskaya nab. 7/9, 199034 Saint Petersburg, Russia
4   Kandalaksha State Nature Reserve, Lineynaya 35, 184042 Kandalaksha, Russia
*   Correspondence: peteryershov@yandex.ru

**Abstract:** The European flounder, *Platichthys flesus*, is a polymorphic flatfish, which has a large population variation in the proportion of left-sided and right-sided morphs across its geographic range. We compared the frequencies of these morphs in the White Sea (Kandalaksha, Onega, Dvina, and Mezen bays), the region in the northeastern part of species' range adjacent to the Arctic. The proportion of the two morphs in the populations of White Sea flounders showed high variability and specific regional characteristics. The highest frequency of left-sided individuals was observed in the northwestern (Kandalaksha Bay) and southwestern (Onega Bay) parts of the White Sea. Flounders living in the eastern part of the White Sea (Dvina and Mezen bays) showed a much lower frequency of this trait. No consistent pattern of geographic variation in the proportion of the morphs was found in the geographic range of *P. flesus*. The lowest frequencies of left-sided individuals were recorded in the flounder populations living at the eastern and western margins of the geographic range. Geographic variation in the proportion of left-sided individuals in flounder populations is likely to be determined by a set of biotic and abiotic factors. Selective influence of the latter, acting through the trophic relationships of this species with other marine organisms, can differ in different parts of flounder's geographic range.

**Keywords:** asymmetry; geographical variation; morph proportions; Pleuronectidae; lateral polymorphism; White Sea

## 1. Introduction

The European flounder *Platichthys flesus* (Linnaeus, 1758) (Pleuronectidae) is a marine and brackish fish distributed in the western part of the Mediterranean, along the Atlantic coast of Europe, around the British Isles and Ireland, in the North, Baltic, Barents and White seas and eastward to the southwestern part of the Novaya Zemlya archipelago and Kara Bay of the Kara Sea [1–4]. Individuals of this species can be either left-sided or right-sided depending on the side of the body on which the eyes lie at the early developmental stages following metamorphosis. In the left-sided (reversed) flounders, both eyes are on the left side of the body, while in the right-sided individuals they lie on the right side. Both morphs are present in populations of this species in varying proportions [5–8]. Crossbreeding studies conducted on the starry flounder *P. stellatus* (Pallas, 1787), a polymorphic congeneric species of *P. flesus*, have shown that body asymmetry direction is under moderate genetic control [9,10].

Studies of lateral polymorphism in the populations of the European flounder are still rather scarce and the majority of information on the frequency of morphs was obtained

from the flounders caught in the Baltic Sea off the coasts of Sweden, Germany and Estonia [5–7,11]. Fornbacke et al. [7] have reported a clinal change in the proportion of left-sided individuals in flounder catches along the coast of Sweden, but the results of other authors on Baltic and North Sea flounders have shown a very high variation in the proportion of morphs in different populations [5,6,8,12]. In the White Sea, the frequency of reversed individuals was measured for some flounder populations from the Kandalaksha (Velikaya Salma Strait), Onega, Dvina, and Mezen bays [13–16]. However, for White Sea flounders, our knowledge of regional variation in this trait remains incomplete because of the paucity of information on the flounder populations inhabiting the western part of the sea, and because of the lack of statistical estimation of the observed variation in this trait among the samples studied by different authors. The study of the phenotypic diversity of flounders inhabiting the edge of the northeastern part of the range, on the boundary of Arctic, is of particular interest with the respect to the adaptive role of the lateral polymorphism of *P. flesus* in extreme habitats. No analysis has yet been conducted to date of the geographic patterns of variation in frequency of lateral morphs in *P. flesus* populations across the species range. Such data are necessary to study various mechanisms of maintaining lateral polymorphism in this species. Yershov et al. [17] have shown that a significant determinant of the frequency of reversed individuals in flounders from the White Sea was the location of the population (factor "Bay"), while other possible factors (length, age, and sex) did not influence the ratio of morphs. Besides that, no statistically significant and consistent changes in interannual variation in the proportion of left-sided individuals were found in flounder populations from the Onega, Dvina and Mezen bays [18]. All these findings allow us to conduct comparative studies of morph proportions in flounder populations on the basis of recent and published data.

The primary goals of the present study were: (1) to analyze spatial variation in the proportion of the left- and right-sided morphs of *P. flesus* in different bays from the White Sea; (2) to identify variation in the proportion of these morphs of *P. flesus* in the geographic range.

## 2. Materials and Methods

Specimens of *Platichthys flesus* for this study were collected in different bays of the White Sea during regular expeditions made by the Polar branch of the Russian Federal Research Institute Of Fisheries and Oceanography (VNIRO) and the Zoological Institute of the Russian Academy of Sciences (May–August 2014–2021). In Kandalaksha Bay, fish were caught in the Chupa Inlet, totaling 584 individuals (Figure 1). In Onega Bay, flounders were collected in two locations: at the head of the bay near Kiy island, situated in the mouth of the Onega River (n = 1186), and in the mouth of the Nyukhcha river (n = 1144). In the Dvina and Mezen bays, fish were caught in the estuarine zones of the Northern Dvina (n = 2613) and Mezen rivers (n = 905). For the generalized comparative analysis, the material collected during the present study was supplemented by data on previous catches made in 2001–2013 in the Onega, Dvina and Mezen bays (collections of G.Fuks and other staff of the SevPINRO, Arkhangelsk [16]). Flounders were collected in rivers and coastal waters at different depths using variable mesh gillnets (mesh size of 30–50 mm) and traps.

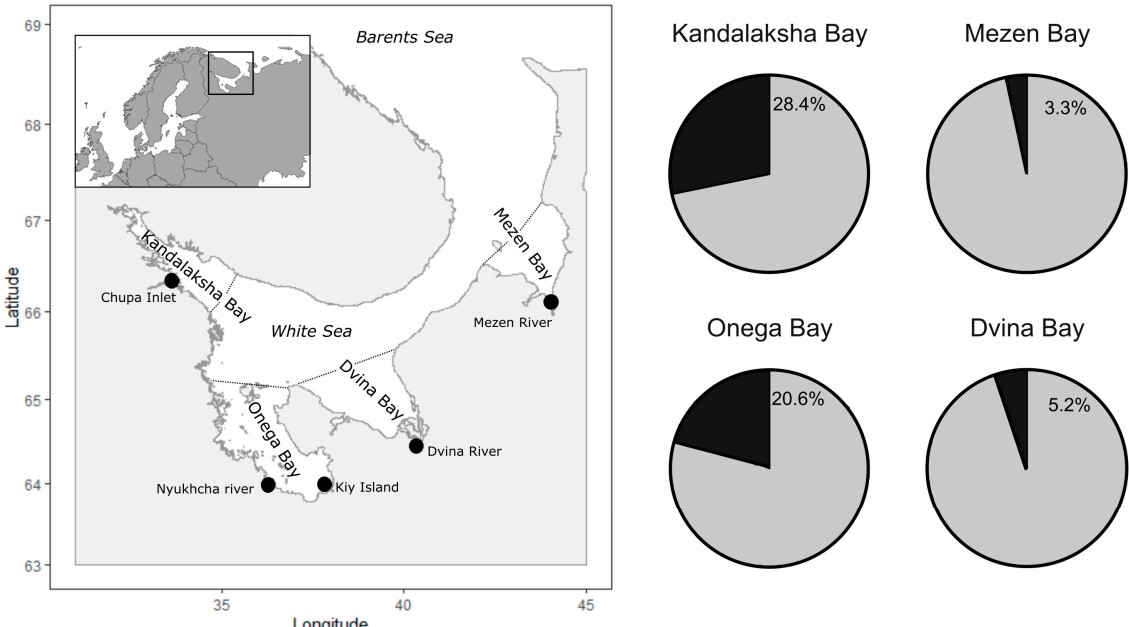

**Figure 1.** Location of sampling areas in the White Sea. Frequency of left-sided *Platichthys flesus* (%, black sectors) morph from the four bays of the White Sea.

The caught fish were examined fresh. Fish were scored for asymmetry morph and the number of left- and right-sided individuals was counted in all samples. Pairwise comparisons of ratios of morphs for flounders caught at different collection sites was performed using χ2 test [19]. Bonferroni's correction was applied to adjust the significance level for multiple comparisons [19]. Because males and females showed no difference in frequency of morphs [16,17], the comparison of samples was performed without differentiating for sex.

We conducted a systematic search for published data that provide information on the proportions of morphs in flounders from different populations. Further, using the data of the Global Biodiversity Information Facility (GBIF.org (accessed on 28 September 2022) GBIF Occurrence Download https://doi.org/10.15468/dl.qe7bf3 )), the coordinates of 5992 unique points of flounder occurrence sites in the European part of the range were obtained. Based on these data, the geometric center of the species range was calculated as a point with geographic coordinates equal to the mean latitude and longitude (54.29339 N, 8.772293 E); then, the distance from each flounder fishing area indicated in the published studies (Table 1) to the calculated geometric center of its range was estimated. To analyze the pattern of geographic distribution, we built two regression models. The first model was built for the material collected in the European part of the flounder range (Table 1, # 14–39), and the second one for sites located in the Arctic region (Table 1, # 1–13). We did not combine both data sets (European and Arctic) into one model, as this would lead to the appearance of collinearity between the predictors. The dependent variable in the models was the proportion of left-sided morphs (it was assumed that the values of this variable correspond to the beta distribution). The distance from the geometric center of the European part of the species range was used as the first predictor in both models. Longitude was considered as a second predictor, additionally characterizing the geographic location. Latitude was not included in the models because it showed collinearity with other predictors.

**Table 1.** Proportions of left-sided *Platichthys flesus* morph in various populations based on the published and original data.

| No | Locality | Sample Size | Left, % | Source of Data |
|----|----------|-------------|---------|----------------|
| 1 | Chupa Inlet, Kandalaksha Bay, White Sea, Russia | 584 | 28.4 | Present study |
| 2 | Delta of the Northern Dvina River, Dvina Bay, White Sea, Russia | 5007 | 5.2 | Present study |
| 3 | Mezen river, Mezen Bay, White Sea, Russia | 2272 | 3.3 | Present study |
| 4 | Nyukhcha river and Kiy island area, Onega Bay, White Sea, Russia | 4527 | 20.6 | Present study |
| 5 | Delta of the Northern Dvina River, White Sea, Russia | 897 | 4 | [15] |
| 6 | Kolezhma river, Onega Bay, White Sea, Russia | 358 | 28.5 | [6] |
| 7 | Kuz Inlet, Onega Bay, White Sea, Russia | 187 | 31.3 | [13] |
| 8 | Delta of the Northern Dvina River, White Sea, Russia | 2394 | 4.7 | [16] |
| 9 | Mezen Bay, White Sea, Russia | 1367 | 3.1 | [16] |
| 10 | Velikaya Salma Strait, Kandalaksha Bay, White Sea, Russia | 957 | 37.1 | [14,15] |
| 11 | Murman coast, Barents Sea, Russia | no data | 39.6 | [20] |
| 12 | Murman coast, Barents Sea, Russia | 475 | 44.5 | [21] |
| 13 | Murman coast, Barents Sea, Russia | 25 | 40 | [22] |
| 14 | Danish Belt Sea | 49 | 25.4 | [8] |
| 15 | Den Helder, North Sea, Netherlands | 75 | 48 | [12] |
| 16 | Eckernforde Bay; Laboe (Kiel), Baltic Sea, Germany | 3331 | 42.7 | [11] |
| 17 | Elbe river mouth, North Sea, Germany | 225 | 23.6 | [5] |
| 18 | English Channel, Plymouth, UK | 1120 | 5.4 | [5] |
| 19 | English Channel, Plymouth, UK | 40 | 7.5 | [8] |
| 20 | Hiiumaa, Baltic Sea, Estonia | 800 | 35 | [6] |
| 21 | Karlskrona, Baltic Sea, Sweden | 631 | 22.4 | [7] |
| 22 | Langeoog, Wadden Sea, North Sea, Germany | 26 | 35 | [22] |
| 23 | Loughor estuary, South Wales, UK | 64 | 4.7 | [8] |
| 24 | Lysekil, Skagerrak, Sweden | 653 | 27 | [7] |
| 25 | Mandjala, Saaremaa, Baltic Sea, Estonia | 200 | 22.5 | [6] |
| 26 | Mevagissey harbour, UK | 192 | 5.7 | [5] |
| 27 | Millport, Cumbrae, Scottish coast, UK | no data | 6.7 | [23] |
| 28 | Neustadt Bay, Baltic Sea, Germany | 90 | 34.4 | [5] |
| 29 | North Bull Island, Dublin Bay, Irish Sea | 590 | 5.6 | [24] |
| 30 | Nynashamn, Baltic Sea, Sweden | 186 | 20.1 | [7] |
| 31 | Oland, Baltic Sea, Sweden | 1673 | 21.1 | [7] |
| 32 | Pudisoo, Baltic Sea, Estonia | 1271 | 33.1 | [6] |
| 33 | Rostock, Baltic Sea, Germany | 15 | 26.7 | [25] |
| 34 | Sorve peninsula, Saaremaa, Baltic Sea, Estonia | 200 | 39.5 | [6] |
| 35 | Strömstad, Skagerrak, Sweden | 455 | 27.5 | [7] |
| 36 | Thames estuary, London, England | 50 | 18 | [8] |
| 37 | Trondheimsfjord, Norway | 269 | 32.3 | [26] |
| 38 | Vastervik, Baltic Sea, Sweden | 193 | 21.2 | [7] |
| 39 | Zuiderzee, North Sea, Netherlands | 50 | 44 | [12] |

Statistical processing was performed using the functions of the statistical programming language R version 4.0.5 [27]. We used the "betareg" package [28] to fit the regression model. To check collinearity of predictors in the models the variance inflation factor was estimated using functions from the "car" package [29]. No collinearity was detected in final versions of the models. Analysis of residuals did not reveal violations of linear models' assumptions for both models.

### 3. Results

The proportion of left-sided individuals in flounder samples from different bays of the White Sea ranged from 3.3% to 28.4% (Figure 1). The highest frequencies of left-sided flounders were recorded in a population from Kandalaksha Bay (Chupa Inlet). The frequencies of reversed individuals in the samples taken from two different places in the southern part of Onega Bay varied from 19.8% (Nyukhcha river area) to 20.9% (Kiy island area) and had equal values ($\chi 2$, $p > 0.05$). For further analysis, the data on flounder catches from these locations in Onega Bay were pooled into a single dataset. The comparison of populations from the northwestern (Chupa Inlet, Kandalaksha Bay) and southwestern (Nyukhcha river, Kiy island area; Onega Bay) parts of the White Sea revealed the difference in morph proportions between these two regions (28.4% and 20.6%, respectively; $\chi 2 = 18.72$, $p < 0.01$). The frequencies of reversed individuals in the populations from the eastern part

of the White Sea (Mezen and Dvina bays) were much lower and ranged from 3.3% to 5.2% (Figure 1). Differences between populations from these bays were statistically significant ($\chi 2$ = 11.89, $p < 0.01$). Left-sided individuals were least frequent in the population from Mezen Bay.

Figure 2 shows the frequencies of reversed individuals in the populations of European flounder from the White Sea and other parts of its geographic distribution. At first, consider the variability of this parameter in flounders from the White Sea. In Kandalaksha Bay, flounders from Chupa Inlet differed significantly from those of the Velikaya Salma Strait ($\chi 2$ = 12.18, $p < 0.01$; Table 1, # 1, 10). Flounders from two close locations at the western coast of Onega Bay, namely Kuz Inlet and the Kolezhma river (Table 1, # 6, 7), were similar in frequency of left-sided morph to one another, and also to the flounders from Chupa Inlet ($\chi 2$, $p > 0.05$). Moreover, frequencies of left-sided individuals in samples from Kuz Inlet and Kolezhma river were significantly higher than in samples taken in the southern part of the same bay, namely the Nyukhcha river and Kiy island area ($\chi 2$ = 22.02, $p < 0.01$; Table 1, # 4). The gradual decline in the frequency of left-sided flounder was observed from the head of Kandalaksha Bay towards the head of Onega Bay (Table 1; # 10, 1, 7, 6, 4). The proportion of non-typical morphs in the populations from the eastern part of the White Sea (Dvina and Mezen bays) was 4–5 times lower as compared to the flounder from the head of Onega Bay.

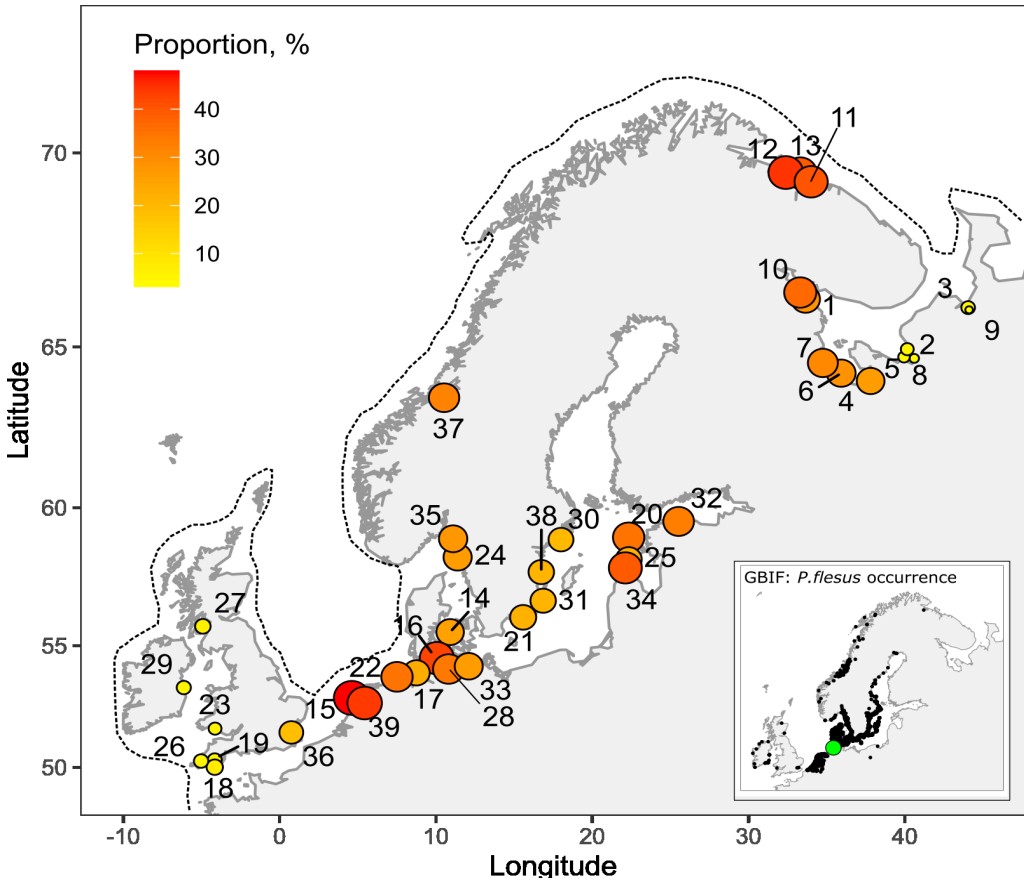

**Figure 2.** Geographical variation in the proportions of left-sided *Platichthys flesus* morph across its range (dotted line). The size of the circles corresponds to the proportion of left-sided individuals in a sample. Data sources are given in Table 2. The small map in the inset shows flounder occurrence according to GBIF. The green dot marks the geometric center of the European part of the species' range.

**Table 2.** Parameters of beta-regression model describing the association of the frequency of left-sided morphs in European populations with distance from the geometric center of European part of the species' range and with geographic longitude.

| Term | Parameter | SE | z-Value | p-Value |
|---|---|---|---|---|
| (Intercept) | −1.043 | 0.1957 | −5.33 | <0.0001 |
| Distance | −0.001 | 0.0003 | −3.86 | <0.0001 |
| Longitude | 0.049 | 0.0109 | 4.55 | <0.0001 |
| Precision coefficient (phi) | 25.049 | 6.8835 | 3.64 | <0.0001 |

The analysis of the data on the frequency of reversed individuals from the flounder's geographic range has shown that all the studied populations were polymorphic and the frequency of left-sided individuals was less than 50% (Table 1). No apparent geographic trend was found for the changes in proportion of left-sided individuals in flounder populations (Figure 2). The highest frequency of left-sided individuals was found for the samples from the Murman Coast of the Barents Sea (44.5%; Table 1, # 12), coastal waters of the Netherlands (48%; #15), and the southwestern (42.7%; German coast, Kiel, Eckernforde; # 16) and eastern (39.5%; Estonian coast; # 34) parts of the Baltic Sea. In addition to the flounder populations from the Dvina and Mezen bays of the White Sea, the lowest frequencies of this trait (4.7–7.5%) were also observed in some areas of the coastal waters of Great Britain and Ireland (Table 1; # 2, 3, 5, 8, 9, 18, 19, 23, 26, 27, 29). These locations are along the marginal area (eastern and western) of the range of *P.flesus*. Parameters of the regression model describing the association of the frequency of left-sided morphs in European populations with distance from the geometric center of the European part of the species' range and with geographic longitude are shown in Table 2. For European populations of flounder (Table 1; # 14–39), a statistically significant relationship was shown between the frequency of left-sided morph with both the distance and the longitude. The negative coefficient (Table 2) indicates a decrease in the frequency of reversed fish with increased distance from the center of the species' range. This decrease is most pronounced when moving from the center of the European part of the range towards the British Isles and Ireland. The positive coefficient for geographic longitude (Table 2) indicates an increase in the frequency of left-sided morph when moving from west to east. For Arctic flounder populations (Table 3, # 1–13), a statistically significant relationship of the dependent variable was shown only with geographic longitude. The negative coefficient (Table 3) indicates a decrease in the frequency of left-sided flounder in the direction from west to east.

**Table 3.** Parameters of the beta-regression model describing the association of the frequency of left-sided morphs in Arctic populations with distance from the geometric center of the European part of the species' range and with geographic longitude.

| Term | Parameter | SE | z-Value | p-Value |
|---|---|---|---|---|
| (Intercept) | 9.166 | 1.7262 | 5.31 | <0.0001 |
| Distance | −0.00003 | 0.0009 | −0.03 | 0.975 |
| Longitude | −0.286 | 0.0348 | −8.22 | <0.0001 |
| Precision coefficient (phi) | 54.5867 | 21.4711 | 2.54 | 0.01101 |

## 4. Discussion

Previously, it was shown that the morph proportions in flounder populations depended on sample location, but not on the size–age and sexual composition of samples [17]. Obtained results demonstrated that the frequency of left-sided flounder differed significantly among four bays of the White Sea. In addition, the comparison between the results of the present study and previously published data [6,13–15] has revealed differences in the proportion of left-sided individuals among flounders living in various parts of the Kandalaksha and Onega bays. This gives evidence that local populations of flounder exist in these two bays. In our opinion, the spatial isolation and morpho–ecological differentia-

tion of flounders in the Kandalaksha and Onega bays of the White Sea is determined to a significant extent by geomorphological and hydrological characteristics of these regions (numerous inlets, rivers with estuarine zones, local cyclonic currents, etc.) which provide a variety of habitats for fish and restrict gene flow between populations along the coast.

Semushin et al. [16] have reported that the proportion of left-sided individuals in the flounder populations inhabiting the White Sea decreases from west to east. Additional data provided by the present study have shown that the left-sided individuals are most common in the northwestern part of the White Sea (Velikaya Salma, Kandalaksha Bay). In Chupa Inlet, located more to the south along the Karelian coast of Kandalaksha Bay, and in Onega Bay (western part, Kuz Inlet and the Kolezhma river), the proportion of left-sided fish in the populations is somewhat lower. An even lower occurrence of left-sided individuals was noted in the southern part of Onega Bay (Nyukhcha river and Kiy island area). A significant decrease in the proportion of left-sided flounders is observed in Dvina Bay, and the lowest proportion of reversed individuals occurs in Mezen Bay. Our results on flounders from the Dvina and Mezen bays were not significantly different from published data [15,16] on frequency of reversed individuals in these populations. Given the quantitative characteristics of this trait, it can be concluded that the frequency of left-sided individuals is much higher in the flounder populations from the Kandalaksha and Onega bays than in those from the Dvina and Mezen bays. It is noteworthy that changes in frequency between these two groups of populations occur abruptly rather than in a gradual fashion. It should be also noted that along the coast of the White Sea, between estuaries of the large Northern Dvina, Onega, and Mezen rivers, the flounder is not numerous and is encountered in small numbers mostly on shoals near estuaries of small rivers.

The total variability of the left-sided morph proportion in flounder populations from the marginal Arctic fluctuated from 3.3% to 44.5% and was comparable with such variability in the European part of the range (4.7–48%). The analysis of regression models, built for the northeastern (Arctic) and European parts of the flounder range, revealed the significant decrease of non-typical *P. flesus* proportion in populations inhabiting marginal (western and eastern) locations of the range. Obviously, the observed decrease in the proportion of left-sided individuals is associated with their reduced survival rate in these locations.

No consistent large-scale geographic trend has been observed for variation in this trait among the studied populations from the coastal waters of Europe (North and Baltic seas). However, Fornbacke et al. [7] found another pattern of geographic variation in the proportion of non-typical morphs for the flounders living in Swedish coastal waters. The frequency of left-sided flounders in the six samples studied decreased gradually from the western coast of Sweden (27.5%), across its southeastern part (22.4%), and then in the northern direction (20.1%). The authors concluded that variation in this trait showed a biogeographic cline along the coast of Sweden. In our opinion, however, this statement calls for further research, because of the uneven sampling across the study area. For instance, this study lacks any data for the coastal waters of the vast part southern coast of Sweden. Numerous data on variation in frequency of reversed individuals in catches from the eastern part of the Baltic Sea (Estonian coast) has been provided by Mikelsaar [6]. The author has not revealed any consistent pattern of variation in this trait for flounders caught in different parts of the study area.

Possible causes producing interpopulation variation in the proportion of morphs in the European flounder remain little explored. Fornbacke et al. [7] have suggested that variation in frequency of left-sided individuals in the samples of flounders along the coast of Sweden is associated with interspecific interactions of young individuals of this species with those of the European plaice *Pleuronectes platessa* Linnaeus, 1758. These authors have argued that food competition of plaice fry on nursery grounds of the Skagerrak Strait is more intense with the right-sided individuals of flounder than with the left-sided individuals. This results in increased survival of the left-sided individuals on the west coast of Sweden, where the numbers of plaice fry in shallow water are high. The authors regarded a larger body size of left-sided fry compared to the right-sided individuals as indirect evidence of the

advantage that the left-sided individuals have in using food resources. However, the results of the present study and previously published data suggest that variation in proportion of morphs in some other parts of flounders' geographic range cannot be explained by the possible influence of competition with the European plaice. The average frequency of reversed individuals in flounders inhabiting the eastern part of the Baltic Sea (Estonian coast), where the plaice is rare [6,30], is much higher than in flounders from the western coastline of Sweden, where the plaice is abundant in shallow water. In the White Sea, the highest frequency of left-sided flounders was recorded in the Kandalaksha and Onega bays, where the plaice are only sporadically caught by fishing gears and their numbers are extremely low [31,32] . It is clear that increased survival of left-sided flounders in these parts of the geographic range is caused by other factors. As regards the differences in size of fry between left- and right-sided individuals [7], they can, in our opinion, be associated not only with differences in their growth rates as a result of the interspecific competition with plaice fry, but also with different hatching times due to the more extended spawning season of flounder; the spawning period of flounder normally lasts about 1 month and its duration varies in different parts of the Baltic Sea [1,30,33,34].

In different parts of flounder's geographic range, the interspecific competition resulting in preferential selection of morphs can also be associated with fish species other than plaice. In the shallow coastal waters of the White Sea, the flounder is known to cohabit with other benthophagous species: the Arctic flounder *Liopsetta glacialis* (Pallas, 1776) and the dab *Limanda limanda* (Linnaeus, 1758) [35,36]. Shatunovsky and Chestnova [37] have shown that in the inlets of Kandalaksha Bay, where the Arctic flounder are relatively abundant, the young European flounder (length< 20 cm) compete with this species for food, and the food spectra of fishes of the two species can overlap by 60–70%. However, information about possible differences in the composition of food consumed by different morphs of the European flounder compared to the Arctic flounder is still lacking. Further studies of dietary habits and behavior in shared nursery areas are needed to evaluate the possible influence of competition with plaice and other species of flatfish on survival of morphs of the European flounder during their first years (0+ and 1+) of life.

Another factor that is likely to influence spatial variation in flounder polymorphism is ecological selection between morphs in individual populations. Russo et al. [24], for example, have revealed intrapopulation differences in food spectra between left- and right-sided individuals among flounders in Dublin Bay (Ireland). The morphs also showed some morphological differences in relative sizes of the premaxillare, the length of the tail peduncle, and the position of eyes, i.e., the characteristics that play a role in targeting and capturing prey items. The authors have argued that the frequency of morphs can be dependent to a certain extent on the characteristics of benthic food resources and the accessibility of certain food organisms, because the left- and right-sided individuals showed preference toward different dietary objects. It remains to be determined to what extent these differences in the composition of prey organisms are important for survival, growth, and fecundity of fish, and whether there are ecological differences between two morphs in various regions of the flounder's geographic range. It should be noted that morphological differences in feeding habits, behavior, and swimming have previously been found between the morphs of the starry flounder *P. stellatus* (Pallas, 1787), which suggested the existence of trophic specialization in the right- and left-sided individuals [38–40].

Fornbacke et al. [7] have noted that variation in proportion of left-sided adult flounders along the coast of Sweden correlates with the salinity gradient of the coastal waters. Over the last two decades, the basin of the Baltic Sea experienced major ecological changes due to eutrophication, intensive fishing, global climate change and other factors [41–43]. Negative trends in ecological health result in the degradation of habitats of many fish species including the European flounder. In the early 2000s, a dramatic fall in numbers and biomass of the European flounder was observed in the northern part of the Baltic Sea [44]. It is clear that, in this situation, the similarity in trends observed for changes in biological traits of fish and certain hydrological data do not necessarily imply any consistent relationship

between these factors in the study area and the observed correlations should be treated with caution. Experimental studies are required to confirm the presumed relationship and determine the direction of natural selection. In our opinion, variations in water temperature and salinity can undoubtedly influence the mortality of flounder in coastal waters, but the main influence of these factors is indirect, i.e., through qualitative changes in those environmental conditions that play a role in successful fish spawning, feeding, and growth.

Future studies should therefore be aimed at the analysis of adaptive strategy and ecological segregation of flounder morphs as a result of their relationships with the environment and competitive interactions with other fish species in different parts of the flounder's geographic range.

In conclusion, the analysis of published and original data on lateral polymorphism across the geographic range of European flounder has demonstrated a high population variation in the proportion of reversed individuals. *P. flesus* did not exhibit large-scale geographic trends in the proportions of left-sided morphs. Judging from the available information, however, the reversed individuals are least frequent in the populations living on the western (Great Britain, Ireland), and northeastern (close to the Arctic region, i.e., the Dvina and Mezen bays, the White Sea) margins of species' geographic range. It should be stressed that populations of *P. flesus* living in the White Sea and other parts of its geographic range are likely to be affected by factors of natural selection specific to each area that influence the proportion of left- and right-sided fish.

**Author Contributions:** All authors contributed towards conceptualization, methodology, analysis, and investigation; P.N.Y. and G.V.F. have made substantial contributions to acquisition of data and conception; V.M.K. has been involved in analysis and interpretation of data; P.N.Y. wrote the original draft of the work. All authors have read and agreed to the published version of the manuscript.

**Funding:** This research received no external funding.

**Institutional Review Board Statement:** Not applicable.

**Data Availability Statement:** Data can be supplied by the corresponding author upon reasonable request.

**Acknowledgments:** We are grateful to the staff of the Coastal Research Laboratory of the Polar branch of FSBI "VNIRO" and White Sea Biological Station "Kartesh" of the Zoological Institute RAS for their help in collecting data during expeditions. This work was carried out as part of the State Task of the Zoological Institute RAS (state registration number no. 122031100283-9).

**Conflicts of Interest:** The authors declare no conflict of interest. The funders had no role in the design of the study, in the collection, analysis, or interpretation of data, in the writing of the manuscript, or in the decision to publish the results.

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
