# Peer review of "Spatial Variation in the Frequency of Left-Sided Morph in European Flounder Platichthys flesus (Linnaeus, 1758) from the Marginal Arctic (the White Sea)"

_diversity, doi:10.3390/d14111004_

Round 1

Reviewer 1 Report

Dear Authors

I hope you are doing fine in these unusual days of the pandemic coronavirus.
I had a chance to review  Manuscript
 no: diversity-1943678
entitled “Spatial variation in the frequency of left-sided morph in European flounder Platichthys flesus (Linnaeus, 1758) from the marginal Arctic (the White Sea)”.

It is an interesting paper dealing with the frequencies of left-sided individuals of a flounder in the White Sea (Kandalaksha, Onega, Dvina, and Mezen bays), the region in the northeastern part of species’ range adjacent to the Arctic. 

In this ms, a proper review of literature especially recently published articles, taxon sampling, and analysis have been performed.

I enjoyed reading the present study which is obviously of great interest to ichthyologists. The objectives are clearly expressed in the introduction, the material and methods are clearly given, and the results are in a good shape.

In my opinion, this manuscript is of interest for publication in Diversity, but not in the present form because it needs major revision.

The comments and corrections are all given in the original pdf file. Some are listed here:

More articles can be included in the introduction section, including the genetic/molecular status of both forms of flounder, laboratory results of the effects of different water parameters on the frequency of both forms, and clinal variations.

The possible clinal trend in TL, SL, weight, morphological characteristics, and sex ratio, … can be added.

The ms requires improvement in English language writing and editing.

Sincerely Yours

Author Response

We would like to thank the referees and the Editor for their positive assessment of the importance and novelty of our work and for their thorough analysis of our manuscript. Their input was carefully considered and their comments were taken into account in the revision. We believe it has helped us to improve the clarity and presentation format of the manuscript. Please find below a list of the revisions made to the manuscript and responses th the referees’ questions and comments.

Reviewer 2 Report

Manuscript “Spatial variation in the frequency of left-sided morph in Euroean flounder Platichthys flesus (linneaeus, 1758) from the marginjal Arctic (the White Sea) by Yershov et al.

The paper deals with interesting observation of divergences in proportions of  left-sided flounders in different locations in the white sea, as well as throughout the range of the species.  The paper is highly descriptive reporting the observed patterns, followed by a detailed discussion, who often are quite speculative. Below are my specific comments on the manuscript.

1 – The methods, especially the statistical methods are greatly lacking. The reader does not know how statistical tests are applied until trying to figure that out by reading the result section. So much more details on sampling and especially statistical analysis are needed.

3 – The authors use really simple statistics for analysing their data. For example using t-test to compare size differences, where (when reading the results) it seems that they were mostly comparing differences between sexes within habitats. For a more detailed information from their size data, giving results on differences between areas and sexes I would strongly suggest the authors to use some kind of linear mixed models for this analysis.

4 – It is impossible to see how the X2 tests were performed. Did the authors considered using Linear Mixed Models for their analysis taking location and sex in to the picture in one, much stronger test.

5 – The methods do not discuss how the authors collected information from the published literature. Was this just a random pick of papers or was there a systematic search. Did the authors not consider statistical analysis on this data?

6 – The discussion is long. It includes quite a lot of speculations regarding the origin of left-right polymorphism in flounder. Often these are highly speculative and the data in the present study does not support those in either way. I would suggest the discussion to be more focused and anchored better in the observed results. Results from other studies on the origin of the polymorphism, could be summarised and combined to a single paragraph.

 6 – Lines 166 – 179. The discussion in this paragraph is highly speculative and are not supported by the data in any way, which is what large part of the later discussion is indicating.  I would suggest this speculations to be taken out of the discussion.

 7 – Discussion in lines 225 – 231 is missing references to support the statements of the authors.

8 – The final statement is highly speculative. The observed patterns do not have to be adaptive through natural selection, as other factors may play an important role, e.g. plasticity and or change.

Author Response

(The authors gave the same response as above.)

Reviewer 3 Report

·      The first sentence of the Introduction is redundant: “The European flounder Platichthys flesus (Pleuronectidae) is a marine and brackish species”. It must be “The European flounder Platichthys flesus is a marine and brackish fish distributed in the western Mediterranean Sea…”

·      I consider the objectives of this work unclear. In one hand, the last paragraph of the Introduction stands “1) to analyse spatial variation in the proportion of the two morphs in flounders from different bays of the White Sea”. It is unclear what do authors imply with “spatial variation in the proportion”. On the other hand, “2) to evaluate large-scale geographic variation in proportion of the left and right-sided morphs in flounder populations across the geographic range of the species. It is unclear what do authors imply by “to evaluate large-scale geographic variation in proportion”. I suggest 1) to compare the proportions of left and right sided morphs of P. flesus in different bays from the White Sea, and 2) to identify variations in the geographic range in the proportions of these morphs in the region based on presence-absence modeling.

·      So, I suggest an alternate title: “Comparisons between left and right-sided morphs in the European flounder Patichthys flesus (Linnaeus, 1758) from the White Sea, northwest coast of Russia as determined by modeling”

·      I suggest editing in the abstract: The European flounder, Platichthys flesus, is a polymorphic flatfish, which has a large population variation in the proportion of left-sided and right-sided morphs across its geographic range. We compared the frequencies of these morphs in the White Sea (Kandalaksha, Onega, Dvina, and Mezen bays). The proportion of the two morphs shown a high variability. The highest frequency of left-sided individuals was observed in the northwestern (Kandalaksha Bay) and southwestern (Onega Bay) White Sea. Flounders eastern part (Dvina and Mezen bays) showed a lower frequency. No consistent pattern of geographic variation in proportion of the morphs was found in the geographic range.

·      Figure 1 is a map. So, I recommend removing “Map showing”. And just “Location of sampling areas in the White Sea.

·      There is a rule violation when reporting results. In lines 170-171, authors failed when mentioning “Our results on flounders from Dvina and Mezen were not significantly different from published data on frequency…” This comparison MUST be done in the Discussion section and it is not part of the results.

·      Again, another violation in lines 185-187 “The analysis of the published data on frequency of the reversed individuals…” This must go in the Discussion.

·      In line 195, it must be “These locations are along the marginal area (eastern and western) of the range of….” Remove “it worth noting.

·      Table 2 and Table 3 are not necessary, and authors just need to mention in text the results from the statistical tests.

·      Indicators in text referring to some numbers, within each table, are not necessary.

Author Response

Authors are grateful to the reviewer for critical remarks which helped us to improve the clarity and presentation format of the MS

Reviewer 4 Report

I suggest putting a small geographic map in figure 1, where it is possible to see which is the macro-area where the samplings were made. For example, a small picture of Europe with a zoom in the sampling area. To allow anyone unfamiliar with the site to understand where it is immediately.

In figure 1, it would be helpful to insert the% symbol in pie charts.

Tables 2 and 3 are not very clear. Add some dividing lines.

The paper "Frequencies of lateral morphs in different age classes of the flounder Platichthys flesus (Pleuronectidae) from the White Sea" is very similar. Still, in the manuscript presented here, there are new analyzes, and they expand the study area.

I found the bibliography dated, with only four papers after 2020.

The introduction leads well to the topic. The materials and methods all seem somewhat confused and poorly explained. This does not help to read the results well. I recommend rewriting more flowing. There is no restriction on the length of the papers.

In discussions, they discuss too much about other papers. Instead, it would be interesting to understand the possible consequences of the manuscript work.

Author Response

We would like to thank the reviewer for the critical remarks and positive assessment of our manuscript

Round 2

Reviewer 1 Report

Dear Authors

Based on my close look at the revised version, it seems that the ms has been improved, and the comments have been implemented.

Hence it can be considered for publication.

However, the journal's formation should be double checked .

Sincerely Yours

Author Response

We would like to thank the reviewer for the positive assessment of our revised manuscript

Reviewer 2 Report

I have examined the reviewed manuscript, who has improved. I still have some concern on the revised MS and how the authors have responded to my former comments. 

1 - I suggested that the authors used linear models for analysing how sex and size had an effect on the proportion. It looks to me that the authors have removed this data from the analysis, thus no linear models on this, and published in a zoological institute proceedings. I think this data would have been an important contribution here and added value to the paper. 

2 - Now the authors have included linear models in relation to something else, which I have a hard time understanding what is.  As far as I can understand from the MS the authors are now testing if the proportion of left vs right proportions is related to the distance from the mean of the distribution of the species, either in the European part of the species range or in the Arctic region. It is hard to see the justification of such a test, and the authors need to explain this further. 

3 - As a follow up on 1. When using data from published literature, the authors need to look in to how meta analysis is performed, reported and analysed. There are special ways of doing so, which the authors have not clearly shown. 

4 - In the reviewed MS it is impossible to see which changes were made in the discussion, as it is now all a track change. However, when reading it through it seems to me to be greatly improved.

Author Response

(The authors gave the same response as above.)

Reviewer 3 Report

Authors did a great job and followed my suggestions. However, I have a last suggestions. If authors are considering conclusions in the last paragraph of the discussion, then they have to remove the last sentence at line 365 and relocate it at the end of the previous paragraph at line 355. So, the last paragraph it will contain conclusions only.

I urge authors to review grammar and style before re-submitting.

Author Response

We would like to thank the reviewer for critical remarks and comments, which helped us to improve manuscript

Reviewer 4 Report

The changes made are sufficient to allow the publication of the manuscript.

Author Response

We would like to thank the reviewer for critical remarks, which helped us to improve the clarity and presentation format of the manuscript

Round 3

Reviewer 2 Report

I have now seen this MS three times. I still have concerns about the proper use of the linear models, but if the authors and the editor believe this to be the right way to analyse the data I will not put my against that. 

Author Response

(The authors gave the same response as above.)
